# The impact of government responses to the COVID-19 pandemic on GDP growth: Does strategy matter?

**Michael König** [1]*, **Adalbert Winkler**[2]

**1** FS-UNEP Collaborating Centre for Climate & Sustainable Energy Finance, Frankfurt School of Finance & Management, Frankfurt, Germany, **2** Centre for Development Finance, Frankfurt School of Finance & Management, Frankfurt, Germany

* m.koenig@fs.de

## Abstract

We analyze whether and to what extent strategies employed by governments to fight the COVID-19 pandemic made a difference for GDP growth developments in 2020. Based on the strength and speed with which governments imposed non-pharmaceutical interventions (NPIs) when confronted with waves of infections we distinguish between countries pursuing an elimination strategy and countries following a suppression / mitigation strategy. For a sample of 44 countries fixed effect panel regression results show that NPI changes conducted by elimination strategy countries had a less severe effect on GDP growth than NPI changes in suppression / mitigation strategy countries: strategy matters. However, this result is sensitive to the countries identified as "elimination countries" and to the sample composition. Moreover, we find that exogenous country characteristics drive the choice of strategy. At the same time our results show that countries successfully applying the elimination strategy achieved better health outcomes than their peers without having to accept lower growth.

## Introduction

The COVID-19 pandemic has caused a dramatic slowdown of the global economy in 2020 [1, 2]. However, there have been substantial differences in GDP developments over time. While Spain recorded a GDP decline of 11.0%, economic activity grew by 2.3% in China [3].

This paper addresses the question whether growth differences are related to the strategy countries adopted in fighting the virus. Our paper is motivated by calls that all countries should pursue a COVID-19 elimination strategy rather than a suppression / mitigation strategy [4, 5], also to lower the economic costs of fighting the pandemic. The calls echo earlier recommendations for the "hammer and dance" approach [6] many governments seem to have followed at the beginning of the first wave. However, high economic costs [1, 2] and the communication challenges associated with the approach [7–10] led many governments to adopt a more gradual approach in line with the suppression / mitigation strategy [11] in the second half of 2020. By doing so they hoped to minimize the economic costs of fighting the pandemic

**Data Availability Statement:** All relevant data are within the paper and its Supporting Information files.

**Funding:** The authors received no specific funding for this work.

**Competing interests:** The authors have declared that no competing interests exist.

and to counter criticism that non-pharmaceutical interventions (NPIs) are excessive. Thus, calls for a widespread use of the elimination strategy conflict with the impression that most countries have switched to a suppression / mitigation strategy.

This paper provides evidence on the impact of COVID-19 strategies pursued by governments of 44 countries on 2020 quarterly GDP growth, i.e. whether from an economic perspective the calls for pursuing an elimination strategy have merit or whether most governments have been right in following a suppression / mitigation strategy in order to minimize output losses. Our benchmark for economic benefits and costs of NPIs is GDP growth only. Other studies make use of concepts such as "value of a statistical life" [12] or "value of production" [13].

We start by providing an analytical framework for categorizing countries in "elimination" and "suppression / mitigation" countries based on their NPI response to rising and falling infection rates and apply this framework to 44 advanced and emerging market countries (see the country list in the S1 File (S1 Table in S1 File)). We continue by analyzing whether exogenous country characteristics make the governments of the sample countries choosing a specific strategy. Finally, we employ fixed effect panel regressions and test whether the strategy choice makes a difference for the economic impact of NPIs, i.e. whether NPIs adopted by elimination strategy countries were associated with a less severe impact on quarterly GDP growth in 2020 than NPIs adopted in suppression / mitigation countries. After some robustness checks, we conclude with a summary.

We find that NPI changes conducted by elimination strategy countries had a less severe effect on GDP growth than NPI changes in suppression / mitigation strategy countries. Thus, a swift and strong response to the virus did not only save lives but also paid off in macroeconomic terms. However, this result is sensitive to the group of countries identified with each strategy and sample composition. Moreover, probit regression results suggest that the choice of strategy was driven by exogenous country characteristics influencing the cost-benefit analysis of each strategy. Thus, we do not claim that running an elimination strategy was causal for minimizing health risks at lower economic costs as it is unclear whether the adoption of such a strategy in other countries would have made NPIs less costly in terms of growth compared to the observed suppression / mitigation strategy outcome. At the same time our results clearly indicate that the elimination countries identified achieved a better economic performance than their non-elimination country peers despite fighting the pandemic by responding swiftly to rising cases with strict NPIs.

## Country strategies in fighting COVID-19

The COVID-19 pandemic was at the centre of government policies in 2020. The Oxford COVID-19 Government Response Tracker (OxCGRT) captures these policies [14] with the Stringency Index serving as the most widely used indicator for assessing the strength of NPIs governments introduce to fight the pandemic within their respective jurisdictions. It shows a rather homogeneous diffusion of interventions in the beginning of the pandemic [15] with most countries following the "hammer and dance" approach [6], i.e. they implemented strict NPIs rather swiftly when fighting the pandemic [16–18]. The experience of countries recording high infection and fatality rates, such as Italy or Spain, indicated that voluntary social distancing is unlikely to be sufficient to keep the pandemic under control [19]. Moreover, governments were convinced that the economic damage associated with strongly rising infection rates, for example due to a loss of working time and rising medical costs [20, 21] or voluntary social distancing [1, 22] would be larger than the costs associated with strong mandatory measures rapidly imposed. This view could draw on historical evidence from pandemics in the

past which exhibited severe negative economic effects [23–25]. Finally, governments hoped that the use of the "hammer" would be rewarded with a long "dance" after the virus has been brought under control.

However, the (economic and social) costs of the "great lockdown" were large [1, 2]. Thus, even though there is cross-sectional evidence that lower fatality rates were conducive to growth in several quarters of 2020 [26, 27], some countries took a more dovish approach when infection rates started to decline and a second wave emerged, i.e. they pursed a suppression / mitigation strategy [11, Table 1].

The strategy aims at allowing countries to "live with the virus" in an as normal way as possible, also in economic terms. Thus, governments respond to rising infection rates at least initially in a measured and gradual way. Strong responses are only foreseen when the number of infections either rise so fast that fatality rates would reach socially non-acceptable levels or that the pandemic gets out of control. When rates fall, the strategy calls for a loosening of measures with large re-open-ing steps to be taken well before the 7-day incidence has approached zero. By contrast, only a few countries, such as Australia, New Zealand, China, and South Korea [11] continued to apply the hammer and dance logic in their COVID-19 policies consistently, i.e. followed an elimination strategy where policies are implemented with the goal of eliminating the virus by swift and strong NPIs whenever infection rates rise above zero and by sustaining these measures until the virus has been basically eliminated from the country (Table 1).

We analyze the strategic choice of governments in 44 countries by assessing the patterns of stringency index and 7-day incidence rates in 2020 given the criteria developed in Table 1 in a quantitative and a qualitative (S2 Table in S1 File) way. Concretely, we identify governments pursuing an elimination strategy if they:

1. respond swiftly and strongly to a rise in infections irrespective of the level they start from;

2. do not ease restrictions when infections rise; and

3. take the largest easing steps when the virus has been basically eliminated only, i.e. at very low, close to zero incidence levels.

**Table 1. COVID-19 strategies–goals and NPI measures when infection rates rise and fall.**

| Strategy | Elimination | Suppression / Mitigation |
|---|---|---|
| Goal | • No community transmission<br>• Eliminating COVID-19 whenever it pops up | • Controlled transmission<br>• Living with COVID-19 but keeping fatality rates at a socially acceptable level / without losing complete control |
| NPIs when infection rates rise | • Maximum action to eliminate community transmission<br>• Swift and strong response when incidence is very low | • Initially measured and gradual response, possibly even lagging, in order to "flatten the curve"<br>• Swift and strong response only when incidence rate reaches levels implying future fatality rates deemed socially unacceptable. Stepwise and target driven approach |
| NPIs when infection rates fall | • Large easing steps only when incidence has reached elimination level (basically zero) | • Immediate (measured) easing steps<br>• Larger reductions substantially before community transmission reaches elimination level |

Source: authors' compilation based on [11]. However, in contrast to [11] we do not distinguish between suppression and mitigation strategies as there is no clear benchmark that separates between (soft) suppression and (hard) mitigation. By contrast, as explained in the main text, this is the case with regard to the elimination strategy on the one hand and the suppression / mitigation strategy on the other hand. Thus, we only distinguish between elimination and suppression / mitigation strategy.

The criteria outlined above imply that an elimination strategy country reports very low 7-day incidences when responding with the strongest rise and with the strongest fall of the stringency index to a given wave of infections. Waves of infections can be defined as resurgences "of the incidence rate [. . .], which cumulatively presents an exponential increase in the number of cases of the disease in a given time period" [28]. Thus, we start our quantitative analysis by identifying for each country waves of infections. In doing so, we are aware that infection and stringency data faces reliability issues in several countries, such as Russia (see https://www.nytimes.com/2020/12/29/world/europe/russia-coronavirus-death-toll.html, accessed online 25 March 2021), Turkey (https://www.ncbi.nlm.nih.gov/pmc/articles/PMC7436880/, accessed online 20 March 2021), or Mexico (https://www.theguardian.com/world/2020/oct/26/mexico-coronavirus-death-toll-much-higher-official-number, accessed online 29 March 2021).

For most countries, wave identification is an easy undertaking. However, in some cases, such as South Africa or Brazil, we observe smaller blips and movements which we do not identify as waves on their own as they occur within larger upward and downward movements of infections. By contrast, we regard even tiny ups and downs as "waves" if they are preceded and followed by rather long periods of basically unchanged, close to zero infections, as it is the case in China and New Zealand. Following this methodology we find that in the course of 2020 some countries, notably in the southern hemisphere, showed only one wave while most countries recorded at least two waves emerging. Accordingly, we limit our analysis of strategic choices to the first wave and the rise of the second wave (for details see S1 Table in S1 File).

We continue by identifying for each country the largest upward jump ($\Delta SI$) in the stringency index (3-day moving average), i.e. the "hammer", when infections rise in the first and the second wave ("lockdown 1" and "lockdown 2" in columns 3 and 11 of S1 Table in S1 File) and the largest drop of the stringency index, i.e. the invitation to "dance", when the first wave recedes from its peak ("re-opening 1", column 7, S1 Table in S1 File). Thus, our assessment picks up on the Response-Risk ratio developed in [14]. However, in contrast to the Response-Risk ratio our approach does not relate the maximum level of governments' responses to the total number of infections but the maximum absolute change in governments' responses to the 7-day incidence rate at this day. Moreover, we also account for governments' behavior when the number of new cases declines, i.e. for the exit from lockdown policies. Finally, we include in our measure the response to the rise in infections in the second wave (if such a wave and the associated response are observed).

*New SI* shows the level of the stringency index after the largest jump or fall of the index. Finally, we record the 7-day incidence (*IR*) at the days of the largest changes of the stringency index (columns 5, 9, 14 in S1 Table in S1 File). The incidence reveals the strategy the respective country pursues as very low incidences indicate that countries follow an elimination strategy by responding swiftly (slowly) and strong when infections rise (fall), while high incidences indicate that the country opts for a suppression / mitigation strategy as its strongest response occurs late (early) when infections rise (fall).

We define countries showing the strongest response in either direction at a 7-day incidence below 5 as countries pursuing an elimination strategy, while countries responding most decisively at an incidence above 5 are referred to as suppression / mitigation countries. We are aware that by setting a common benchmark based on reported cases, there is a risk that countries might qualify as elimination countries because they test less extensively than other countries and hence report fewer cases. Indeed, cross-country data on COVID-19 testing [29] point in this direction as countries reporting incidence rates below 5 test significantly less than their peers. However, these countries also show a much lower test positive rate than their peers, consistently below the 5% benchmark set by the WHO for categorizing countries as having the

**Table 2. Government responses to the pandemic.**

| | | Wave 1 | | | | | | | Wave 2 | | | |
| | | Lockdown 1 | | | Re-Opening 1 | | | | Lockdown 2 | | | Index |
| | # of Waves | ΔSI | new SI | IR | ΔSI | new SI | IR | Mean IR (Wave 1) | ΔSI | new SI | IR | Mean IR (Total) |
| | | (1) | (2) | (3) | (4) | (5) | (6) | (7) | (8) | (9) | (10) | (11) |
| Mean | | 41.90 | 65.46 | 7.36 | -15.08 | 53.02 | 19.79 | 13.58 | 16.65 | 64.59 | 190.30 | 68.21 |
| Median | 2.00 | 42.59 | 67.75 | 2.42 | -12.50 | 51.39 | 5.65 | 8.14 | 15.51 | 66.66 | 134.57 | 54.19 |
| SD | | 10.21 | 16.08 | 11.82 | 9.44 | 15.20 | 37.66 | 18.57 | 9.83 | 12.63 | 206.44 | 67.27 |
| Max | 3.00 | 63.89 | 91.36 | 67.42 | -1.85 | 81.94 | 157.63 | 78.92 | 46.76 | 82.41 | 810.15 | 272.00 |
| Min | 1.00 | 16.66 | 19.44 | 0.01 | -43.52 | 26.85 | 0.02 | 0.09 | 3.70 | 34.26 | 0.00 | 0.07 |

Note: *ΔSI* shows the maximum change (absolute difference) in the stringency index (3-day moving average) within one week and *new SI* the stringency level reached after the maximum hike / reduction of NPIs. *IR* (incidence rate) informs about the number of officially reported infections per 100,000 inhabitants over the 7-day period prior to the maximum change in *SI. Mean IR (Wave 1)* is the mean value of IR in *Lockdown 1* and *Re-Opening 1. Mean IR (Total)* also includes values of IR for *Lockdown 2*. The lower the IR value the earlier (later) a government enacted maximum changes in the stringency index when facing a rise (decline) in the 7-day incidence. Reading example: For Lockdown 1, the average largest jump (41.9) happened at the time when the incidence rate over the last 7-days was 7.36 per 100,000 inhabitants. Source: authors' calculations. Numbers based on the country sample of 44 countries. For further details see S1 Table in S1 File.

pandemic under control. Moreover, when replacing the 7-day incidence rate with a 7-day moving average for COVID-19 related deaths we identify exactly the same countries as elimination strategy countries even though the correlation between intensity of COVID-19 tests and related deaths is substantially smaller than between COVID-19 tests and number of cases. Thus, we are confident that differences in testing density across countries do not drive our results. We thank an anonymous reviewer for alerting us to this issue.

For the countries of our sample as a whole results show that in the beginning of the pandemic policies in line with the elimination strategy were quite popular. The median 7-day incidence at the day with the strongest rise in the stringency index is 2.42 (mean: 7.36, standard deviation (SD) 11.82 –Table 2, column 3). 26 (16) countries record the largest increase in the stringency index at 7-day incidences below 5 (1). Thus, many countries made use of the "hammer" early. Only five countries, namely Iceland, Switzerland, Germany, Luxembourg and Norway, acted most decisively at a rather late stage, i.e. when the 7-day incidence was above 20. Thus, these countries indicated from the very beginning of the pandemic that they aim at following a rather gradual approach when responding to rising infections in line with a suppression / mitigation strategy. The remaining countries were somewhat in the middle.

Developments after the peak of the first wave offer the second opportunity to identify the COVID-19 strategy countries have been pursuing in 2020. Results show that the elimination strategy remained popular with 20 (11) countries displaying behavior consistent with this strategy by taking the largest step of re-opening when the 7-day incidence was below 5 (1). At the same time, the median 7-day incidence at the day of the largest decline of the stringency index is 5.65 (Table 2, column 6), which is substantially larger than at the day of the largest rise. This suggests that the suppression / mitigation strategy gained some support after the peak of the first wave as several countries responded relatively early to falling infection rates with large easing steps. Nine countries show the strongest easing of NPIs at a 7-day incidence above 20. For the first wave as a whole (Table 2, column 7), the median value of the 7-day incidence amounts to 8.14. Moreover, less than 40% of the countries pursued an elimination strategy consistently throughout the first wave by showing an average 7-day incidence at the days of the strongest rise and fall of the stringency index below 5.

A major change in governments' NPI responses is observed when the second wave emerges. Now, the strongest NPI action occurs at elevated 7-day incidences (Table 2, column 10) with a

median of 134.57 (mean: 190.30, SD: 206.44). This is dramatically higher than in the first wave indicating that governments might have become aware of a "lockdown fatigue" [30]. Thus, most countries follow a suppression / mitigation strategy after the first wave while only six out of 38 countries continue to follow an elimination strategy, namely China, South Korea and New Zealand with the largest increase in the stringency index at a 7-day incidence rate below 1, and Australia, Spain and Japan at an incidence level below 5. Note, that there is either no second wave of infections and / or no second rise in the stringency index for Indonesia, India, Chile, Columbia and Argentina.

We summarize the evidence from lockdowns 1 and 2 and the reopening in the first wave by calculating the mean 7-day incidence recorded at the days of the strongest rise and fall of the stringency index for each country. The sample median is 54.19 (mean: 68.21, SD: 67.27 – Table 2, column 11) with only China, South Korea, Japan, Australia, and New Zealand, clearly and consistently following an elimination strategy with mean 7-day incidence values between 0 and 1.6 (S1 Table in S1 File). Spain, Turkey and Indonesia follow with values above 7, i.e. almost five times larger than for New Zealand, the country with the highest value in the elimination group. Mean values of the remaining 35 countries indicate that they have not consistently followed the elimination strategy in 2020.

The analysis run above might not fully capture the complexity of the interactions between infection rates and government NPI responses which are at the heart of the strategy definitions. Against this background, we also perform a qualitative analysis (S2 Table in S1 File). Concretely, we make use of the data provided by Oxford University [14] and plot for each country the 7-day incidence rate and the 3 day moving average of the stringency in 2020. Based on these plots we identify non-elimination countries by searching for episodes where governments did not tighten or even eased NPIs despite rising incidence rates, i.e. conducted NPI policies clearly inconsistent with the elimination strategy. Results are largely in line with those observed in the quantitative analysis. In particular, they again clearly identify China, South Korea, Japan, Australia and New Zealand as countries which followed the elimination strategy. However, with some qualifications, the qualitative approach suggests that Argentina, Chile and Mexico could be categorized as countries running an elimination strategy which failed: while NPI policies were not openly inconsistent with elimination strategy principles, incidence rates did not remain at levels close to zero. We follow up on this result by performing some robustness checks with an extended group of elimination countries including Argentina, Chile and Mexico.

## The endogeneity of COVID-19 strategies

### Statistical analysis

Our finding that only a few governments followed the elimination strategy consistently during 2020 is in line with observations made in [11]. However, it represents a puzzle if the elimination strategy were to represent the best approach for addressing the COVID-19 pandemic from a health and an economic perspective. Assuming that governments do not act irrationally this puzzle might be solved by accounting for exogenous, local conditions which facilitate a successful application of the elimination strategy in the five countries in terms of effectiveness and costs.

To this end, we move from a descriptive analysis to probit regressions and test whether the adoption of an elimination strategy was influenced by such conditions. We start with factors strengthening the effectiveness of the elimination strategy, such as (1) border management "with closely supervised quarantine of all arrivals", and (2) experience with other, notably the SARS pandemics which has institutionalized a "vigorous response" to COVID-19 [11, on the

latter see also [31]. We capture border management by the dummy variable "island country" (*Island*) as border management is greatly facilitated when a country can only be reached by boat or plane even though contagion from abroad remains a possibility [32]. Moreover, many island countries are more remote from other countries than countries with a land border which also facilitates efforts keeping COVID-19 infections at bay. A second dummy variable (*SARS*) with the value of 1 is assigned to all countries with at least one case of SARS-COV-1 infections [33]. We expect that island countries and countries with a SARS experience will show a higher likelihood of adopting an elimination strategy.

We also account for government effectiveness (*Government*). Theoretically, it could influence the strategic choice in either way. On the one hand, higher government effectiveness might raise the probability of adopting an elimination strategy as only an effective government can implement "decisive actions" swiftly and successfully. Moreover, an effective government is needed to trace transmission chains at very low numbers of infections [34]. On the other hand, higher effectiveness might lower the probability of adopting the elimination strategy as governments of countries with higher management capabilities might be more tempted to believe that they can carefully manage the COVID-19 crisis by "smart" fine-tuning measures in line with the suppression / mitigation strategy. We also run regressions replacing the government effectiveness index by he Global Health Security Index (https://www.ghs¬index.¬org/) as reported for 2019 in order to focus more on the quality of health systems and the preparedness of countries in dealing with a pandemic. However, the index has no explanatory power. This is in line with other studies [35] indicating that the pre-pandemic quality of the health system, as measured by the GHS index, has no impact on the health performance of countries during the pandemic.

Finally, we add *Trade* openness to the list of exogenous factors explaining the choice of strategy, i.e. the sum of exports and imports divided by GDP in 2018. Countries showing a lower degree of trade openness are likely to be more inclined to follow the elimination strategy than highly open economies as the former face lower costs than the latter when cutting ties with other countries as recommended by the elimination strategy given that they can rely on a large internal market. Iceland, despite being an island, allegedly rejected the elimination strategy for the fact that its internal market is too small to sustain the strategy economically [36]. Thus, we expect a negative coefficient for *Trade* (descriptive statistics of the variables employed are presented in S3 Table in S1 File).

## Results

Results (Table 3) show that island status significantly raises the probability of observing an elimination strategy. On average, island countries are between 12 and almost 18 percent more likely to run an elimination strategy than non-island countries. By contrast, the SARS dummy fails to be significant. The same holds for government effectiveness and trade if the SARS dummy is included. However, when dropping the SARS experience from the list of explanatory variables, there is a weakly significant positive effect for effectiveness and a negative one for trade. The latter implies that an infinitesimal small rise in trade openness lowers the probability of countries adopting an elimination strategy by about 0.3 percent. Given large differences in trade openness–for example New Zealand has a ratio of about 56 percent while Switzerland's amounts to 120 percent–results suggest that the New Zealand government might have opted against the elimination strategy if the country were located in the midst of Europe and its economy highly dependent on trade.

Results are robust to an inclusion of Spain, Turkey and Indonesia to the list of "elimination countries" (S5 Table in S1 File). However, when including Argentina, Chile and Mexico, i.e.

**Table 3. Probit regression (average marginal effects).**

| Dependent variable: *Elimination Strategy* | (1) | (2) | (3) | (4) |
|---|---|---|---|---|
| SARS | 0.124 | 0.128 | 0.100 | |
| | (0.094) | (0.092) | (0.079) | |
| | [-0.060; 0.308] | [-0.051; 0.308] | [-0.055; 0.255] | |
| Island | 0.177** | 0.167** | 0.119* | 0.130** |
| | (0.083) | (0.080) | (0.067) | (0.065) |
| | [0.014; 0.340] | [0.011; 0.324] | [-0.012; 0.250] | [0.002; 0.257] |
| Government | | 0.034 | 0.076 | 0.083* |
| | | (0.053) | (0.048) | (0.043) |
| | | [-0.069; 0.138] | [-0.019; 0.171] | [-0.002; 0.168] |
| Trade | | | -0.003 | -0.003* |
| | | | (0.002) | (0.002) |
| | | | [-0.006; 0.001] | [-0.007; 0.001] |
| Countries | 44 | 44 | 44 | 44 |
| Pseudo R$^2$ | 0.26 | 0.27 | 0.40 | 0.36 |
| Chi-2 (p-Value) | 0.02 | 0.04 | 0.01 | 0.01 |

Note: Binary Probit model.

*,**,*** denote significance at 10, 5, and 1 percent levels, respectively. Values represent marginal effects (dy/dx). Values in parenthesis represent robust standard errors; 95%-confidence interval of dy/dx is in squared brackets. For robust probit regression results showing estimator coefficients in the form of $\Pr(y_i) = \alpha + \sum_{j=1}^{J} \beta_{j,i} + \varepsilon_i$ see S4 Table in S1 File).

the countries which under the qualitative analysis might also be categorized as elimination countries, only *Trade* remains significant (S5 Table in S1 File). By contrast, *Trade* becomes insignificant when limiting the analysis to countries with a population of more than 3 million (the test is is motivated by the consideration that very small countries might face a higher hurdle for qualifying as an elimination strategy country as–given small population size–a comparatively low absolute number of infections might imply relatively high 7-day incidences to which authorities are unable to respond quickly enough to be categorized as an elimination country under the quantitative approach). As trade openness correlates negatively with population size, the finding indicates that very small, non-island countries are candidates for adopting a suppression / mitigation strategy. We continue by replacing the island dummy with a dummy taking the value 1 if a country has not more than two land border neighbours (*Two Neighbour Max*). Results (S5 Table in S1 File) show that the coefficient of the new dummy is also positive and significant. This confirms the importance of border management issues for the COVID-19 strategy choice. Finally, the baseline is supported when running a logit regression (S5 Table in S1 File). We conclude from this that the choice of the COVID-19 strategy reflects at least partly exogenous characteristics which influence the relative effectiveness and costs of the strategies considered. We now turn to the question if the elimination countries possibly exploiting favorable implementation conditions benefitted in terms of growth compared to their suppression / mitigation strategy peers.

## The growth impact of NPIs under different strategies

### Statistical analysis

Our analysis of the growth impact of NPIs under different strategies follows the methodology employed by [24] when assessing the economic damage associated with fatality rates of the

Spanish influenza at the end of the 1910s for 48 countries. Concretely, we run panel fixed effects regressions in order to mitigate omitted variable bias and endogeneity concerns when analyzing the impact of NPIs on economic activity [2, 37] by controlling for time invariant country characteristics. Examples for time invariant characteristics which might impact the growth performance under COVID-19 conditions are the degree of integration into the global economy via contact-intensive sectors like tourism, and GDP per capita as the latter influences a country's ability to conduct economic policies fighting the COVID-19 recession [26, 27]. The observation period begins 2014 in order to exclude the effects of the global financial and euro crisis on GDP developments. Accordingly, we estimate the following time fixed effects panel regression for the period from 2014 Q1 to 2020 Q4:

$$\Delta y_{i,t} = \alpha + \beta_1 \text{COVID}_{i,t} + \beta_2 \text{Strategy} * \text{SI}_{i,t} + \beta_3 \text{Share}_{i,t} + \varepsilon_i. \qquad (1)$$

where $\Delta y_{i,t}$ is the quarterly growth rate (*Growth*) of real GDP in country *i* at time *t*, i.e. the change in real GDP in percent over the same quarter in the previous year. $\text{COVID}_{i,t}$ represent the stringency index (*Stringency*) and the fatality rate (*Fatality*) calculated as the number of confirmed deaths related to COVID-19 per 100,000 inhabitants of country *i* at time *t*. They are zero for all quarters until 2020 Q1 and are supposed to capture the impact of mandatory and voluntary social distancing on economic activity [38]. We opt for the fatality rate rather than the rate of infections as the latter is allegedly subject to larger cross-country differences unrelated to health risks triggered by COVID-19, such as different testing and reporting policies, than the former. We expect that a rising stringency index and a rising fatality rate are associated with a decline in domestic economic activity as the former captures stronger mandatory and the latter stronger voluntary distancing. Both forms of social distancing negatively influence GDP growth.

We continue by including the stringency index lagged by one quarter to account for time lagging effects (*Stringency (lag)*). Thus, we explore whether countries opting for a strong response to the pandemic in the previous quarter were rewarded with a higher growth rate in the current quarter as suggested by the "hammer and dance" effect. Thus, we expect a positive coefficient. Finally, we add an interaction term between a dummy variable for countries following the elimination strategy and the stringency index (*Elimination x Stringency*). With this variable we test whether changes in the stringency index over time had a different influence on growth in elimination countries than in countries which at some point in the course of 2020 opted for a suppression / mitigation strategy. Claims that an elimination strategy is also the better strategy in economic terms suggest a significant positive coefficient while the view that an elimination strategy has unacceptable economic costs implies a negative coefficient.

Economic activity over time is not only affected by the COVID-19 variables. Before Q1 2020 GDP growth was determined by global and local developments unrelated to COVID-19. Time fixed effects capture global developments. In addition, we account for time variant local developments by including the deflated share price index ($Share_{i,t}$) as an independent variable. The variable is supposed to reflect the stance of economic policies pursued in the countries of our sample as well as other local time variant factors influencing GDP growth before and during the pandemic.

Descriptive statistics (Table 4) provide a first indication that strategy might matter for growth as the elimination countries show a significantly better growth performance than their suppression / mitigation strategy peers despite recording a stringency index which on average is only marginally and insignificantly lower than in the suppression / mitigation strategy countries. By contrast, they record a significantly lower fatality rate (see also S7 Table in S1 File).

**Table 4. Descriptive statistics for 2020 Q1-Q4.**

| Variable | Total sample | | | | | Suppression / mitigation Country | | | | | Elimination countries | | | |
|---|---|---|---|---|---|---|---|---|---|---|---|---|---|---|
| | **Mean** | **Median** | **SD** | **Obs.** | | **Mean** | **Median** | **SD** | **Obs.** | | **Mean** | **Median** | **SD** | **Obs.** |
| Growth (%) | -4.37 | -3.35 | 5.56 | 176 | | -4.70 | -3.51 | 5.730 | 156 | | -1.78 | -0.99 | 4.55 | 20 |
| Stringency | 51.12 | 57.40 | 21.99 | 176 | | 51.47 | 57.81 | 22.42 | 156 | | 48.37 | 53.56 | 18.56 | 20 |
| Stringency (lag) | 35.76 | 32.72 | 29.11 | 176 | | 35.80 | 27.79 | 29.48 | 156 | | 35.49 | 34.64 | 26.68 | 20 |
| Fatality | 14.88 | 3.72 | 22.37 | 176 | | 16.73 | 4.77 | 23.12 | 156 | | 0.44 | 0.23 | 0.73 | 20 |
| Share | 1.07 | 1.04 | 0.29 | 176 | | 1.07 | 1.04 | 0.29 | 156 | | 1.08 | 1.01 | 0.27 | 20 |

Note: Total sample (n = 44 countries for 4 quarters = 176 Obs.), Elimination countries (n = 5), Suppression / mitigation countries (n = 39). *Growth* and *Share* are taken from / calculated based on data from OECD. For Argentina we calculate the *Share* price based on data from the Federal Reserve Bank of St. Louis (CPI Index) and the S&P Mervel Index. COVID-19 variables are taken from Oxford University [14]. *Stringency* is always based and calculated using the 3-day moving average. *Stringency (lag)* is zero in Q1 2020. Descriptive statistics for *Growth* and *Share* for the period Q1 2014 –Q4 2020 are presented in S6 Table in S1 File.

## Results

Results of the time fixed effects panel regression (Table 5) show that a rising stringency index substantially lowers GDP growth in the respective quarter. Concretely, a rise of the stringency index by one point lowers GDP growth by 0.09 percentage points (Table 5, column 1). Moreover, changes in the fatality rate have a significantly negative influence on GDP growth developments. Thus, the results of the first specification provide broad support for the view underlying the suppression / mitigation strategy as they highlight the negative economic effects of fighting the COVID-19 pandemic via NPIs. At the same time, however, a strict and swift rise in NPIs might still be beneficial if–as it is the case in elimination countries (Table 4)– such a rise is associated with a very low or close to zero fatality rate.

We continue by including the stringency index lagged by one quarter. The significant and positive coefficient shows that tougher NPIs adopted in the previous quarter are associated with a positive effect on GDP growth in the current quarter. Moreover, the effect is sizeable as a rise in the stringency index by one point in the previous quarter is associated with a GDP growth rate which is–everything else equal–about 0.08 percentage point higher (Table 5, column 2). This supports the "hammer and dance" view of NPI policies. A strong response to the pandemic in the current quarter has substantial economic costs in the current quarter but is rewarded by a higher growth rate in the next quarter. At the same time, the fatality rate coefficient turns insignificant.

Specification 3 (Table 5, column 3) includes the strategy interaction term which directly tests whether changes in the stringency index in those countries identified as pursuing an elimination strategy (Australia, China, Japan, New Zealand, and South Korea) had a different impact on quarterly GDP growth than in countries with a suppression / mitigation strategy. Results show that this is the case as the coefficient is positive and significant. This implies that the five elimination countries were able to limit the economic damage associated with tougher NPIs compared to the remaining countries identified as countries pursuing a suppression / mitigation strategy. The size of the coefficient suggests that the negative growth effects of a rising stringency index were in the magnitude of one third lower for countries pursuing an elimination strategy than for countries following a suppression / mitigation strategy. Concretely, a rise of the stringency index by one point lowered quarterly GDP growth by 0.08 percentage points in the former, but by 0.12 percentage points in the latter countries. As elimination and suppression / mitigation strategy countries on average implemented NPIs with a similar degree of restrictiveness, this implies that elimination countries conducted NPIs in a way that led to a close to zero fatality rate and less economic damage than in suppression / mitigation

**Table 5. GDP growth, the stringency index and COVID-19 fatality—fixed effects regressions.**

| Dependent variable: *Growth (%)* | (1) | (2) | (3) |
|---|---|---|---|
| Stringency | -0.09** | -0.12*** | -0.12*** |
|  | [0.03] | [0.04] | [0.04] |
| Stringency (lag) | - | 0.08** | 0.08*** |
|  | - | [0.03] | [0.03] |
| Elimination x Stringency | - | - | 0.04** |
|  | - | - | [0.02] |
| Fatality | -0.03* | -0.02 | -0.01 |
|  | [0.02] | [0.02] | [0.02] |
| Share | 2.35*** | 2.48*** | 2.42*** |
|  | [0.74] | [0.74] | [0.75] |
| 2020 Q1 | -0.84 | -0.20 | -0.24 |
|  | [0.81] | [0.81] | [0.80] |
| 2020 Q2 | -6.15*** | -5.67*** | -5.64** |
|  | [2.07] | [2.08] | [2.21] |
| 2020 Q3 | -0.33 | -4.05* | -3.93* |
|  | [1.82] | [2.32] | [2.15] |
| 2020 Q4 | 1.83 | -0.87 | -0.97 |
|  | [1.95] | [2.17] | [2.07] |
| Constant | -0.62 | -0.76 | -0.69 |
|  | [0.85] | [0.85] | [0.87] |
| Time Fixed Effects (TFE) | Q1 2014 –Q4 2020 (pre-2020 quarters not displayed) | | |
| Model | FE | FE | FE |
| Countries | 44 | 44 | 44 |
| R2 (adj.) | 0.70 | 0.71 | 0.71 |
| R2 (within) | 0.71 | 0.71 | 0.72 |
| R2 (overall) | 0.53 | 0.53 | 0.54 |
| R2 (between) | 0.02 | 0.04 | 0.02 |
| Rho (inter. cor.) | 0.52 | 0.53 | 0.53 |
| F-Statistic | 154.28 | 146.10 | 212.57 |

Robust standard errors.

*, **, *** denote significance at 10, 5, and 1 percent levels, respectively. *Stringency* is the Oxford University Stringency Index mean value in the respective quarter. *Fatality* is the number of COVID-19 deaths per 100,000 inhabitants in the respective quarter. Observation period begins in 2014 Q1, *Stringency* and *Fatality* is equal to zero until 2020 Q1. Further notes see Table 4.

countries. Thus, the result supports the view that the countries conducting the elimination strategy in 2020 were able to achieve better health and economic outcomes as a swift tightening of NPIs is associated with a much lower fatality rate and less detrimental growth effects.

For the remaining time-variant variable, the deflated share price index ($Share_{i,t}$), we find the expected positive effect, i.e. rising share prices are associated with rising quarterly GDP growth. Finally, the time fixed effects for the four quarters of the COVID-19 period–we refrain from reporting the time fixed effects for the period 2014–2019 for space reasons–reveal that there is a significant and strongly negative effect for the second and third quarter. We interpret this as the global COVID-19 effect, i.e. GDP growth of any country in our sample would have declined by about six percent during the great lockdown in the second quarter, even if it had not recorded any NPIs, COVID-19 deaths and changes in share prices. No country would

have grown at an unchanged pace in a global pandemic even if it had not been affected by the virus itself.

Overall, results suggest that strategy matters. With the caveat that the strategy choice might have driven by exogenous country factors, we find that elimination countries were able to avoid some of the economic damage associated with fighting the pandemic via changes in NPIs in non-elimination countries. Moreover, for all countries results indicate that a tightening of NPIs implies lower growth in the current quarter as suggested by the "hammer" analogy, but tougher NPIs in the previous quarter raise growth in the current period in line with the "dance" promise.

## Robustness checks

We run a series of robustness checks focusing on the interaction term in specification 3 of the baseline. We start (Table 6, columns 1–4) by testing whether the significance of the interaction dummy is driven by the very exogenous characteristics listed in section 3 as explanatory factors for the choice of strategy. To this end we introduce interaction terms between stringency index and a) island country status (*Island x Stringency*), b) SARS experience (*SARS x Stringency*), c) countries with a maximum of two neighbours (*Two Neighbours Max x Stringency*) and d) countries with an above median trade exposure in the country sample *(High Trade x Stringency)*. Results show that all interaction terms fail to be significant.

This suggests that the exogenous characteristics as such do not significantly influence the growth impact of changes in NPIs. We interpret this as evidence that the positive interaction term in the baseline indicates that strategy matters: island country status alone does not mitigate the negative growth impact of tighter NPIs while the decision to employ an elimination strategy–possibly facilitated by island country status–does so. At the same time, robustness checks reveal that the significance of the interaction term is sensitive to the countries identified as countries pursuing an elimination strategy. Adding Spain, Turkey and Indonesia to the group of countries which pursued such a strategy the coefficient turns insignificant (Table 6, column 5). The same holds when adding Argentina, Chile and Mexico, i.e. countries which pursued such a strategy but failed in terms of implementation–as suggested by the qualitative analysis presented in S2 Table in S1 File.

We also find that the results are sensitive to the sample's country composition. When focusing on OCED members only (Table 6, column 6), i.e. when excluding emerging markets from the analysis, the coefficients of the interaction term and the lagged stringency variable turn insignificant, while the fatality rate becomes significant again. Thus, the growth benefits of pursuing an elimination strategy become indirectly visible via the fatality rate as these countries record close to zero fatality rates. By contrast, some of the suppression / mitigation countries, many of them small, landlocked countries like Switzerland, Hungary, the Czech Republic and Slovenia, are associated with additional growth declines between 1.8 and 3.6 percent due to rising fatality rates. This is confirmed when running OECD specifications without the fatality rate. Then the lagged stringency variable and the elimination strategy interaction term continue to be significant.

Robustness checks also indicate that China is an important driver of the significantly positive interaction term in the baseline. When we exclude China from the analysis, i.e. when focusing on a sample of 43 countries only and with Australia, Japan, South Korea, and New Zealand forming the elimination strategy group (Table 6, column 7), the interaction term turns insignificant. This is not the case when excluding one of the other elimination strategy countries from the sample (S8 Table in S1 File). Finally, we find that the interaction term remains significant when excluding small countries from the sample (Table 6, column 8).

**Table 6. Robustness checks.**

| Dependent Variable: *Growth (%)* | (1) | (2) | (3) | (4) | (5) | (6) | (7) | (8) |
|---|---|---|---|---|---|---|---|---|
| Stringency | -0.12*** | -0.11*** | -0.12*** | -0.12*** | -0.12*** | -0.08*** | -0.10** | -0.13*** |
|  | [0.04] | [0.03] | [0.04] | [0.04] | [0.04] | [0.02] | [0.04] | [0.04] |
| Stringency (lag) | 0.08** | 0.08*** | 0.08** | 0.09*** | 0.08** | 0.04 | 0.05** | 0.08*** |
|  | [0.03] | [0.03] | [0.03] | [0.03] | [0.03] | [0.02] | [0.02] | [0.03] |
| Fatality | -0.02 | -0.02 | -0.02 | -0.02 | -0.01 | -0.03** | -0.01 | -0.02 |
|  | [0.02] | [0.02] | [0.02] | [0.02] | [0.02] | [0.01] | [0.02] | [0.02] |
| *Dummy* x Stringency | -0.01 | -0.01 | 0.01 | 0.01 | 0.02 | 0.02 | 0.03 | 0.03* |
|  | [0.03] | [0.02] | [0.02] | [0.02] | [0.03] | [0.02] | [0.02] | [0.02] |
| Share | 2.48*** | 2.55*** | 2.44*** | 2.46*** | 2.52*** | 1.97*** | 2.47*** | 2.88*** |
|  | [0.74] | [0.75] | [0.76] | [0.75] | [0.75] | [0.63] | [0.77] | [0.93] |
| 2020 Q1 | -0.19 | -0.26 | -0.23 | -0.35 | -0.21 | -1.05* | -0.43 | 0.17 |
|  | [0.80] | [0.78] | [0.79] | [0.79] | [0.80] | [0.56] | [0.83] | [0.85] |
| 2020 Q2 | -5.60*** | -5.96*** | -5.79*** | -6.27*** | -5.60** | -7.66*** | -6.52*** | -5.18** |
|  | [2.04] | [1.97] | [2.00] | [2.09] | [2.13] | [1.53] | [2.18] | [2.41] |
| 2020 Q3 | -3.95* | -4.48** | -4.23* | -4.86** | -3.92* | -3.62* | -3.31 | -3.47 |
|  | [2.24] | [2.10] | [2.23] | [2.30] | [2.22] | [1.88] | [2.27] | [2.31] |
| 2020 Q4 | -0.74 | -1.24 | -1.04 | -1.59 | -0.90 | -0.76 | -0.70 | -0.45 |
|  | [2.11] | [1.93] | [2.12] | [2.16] | [2.10] | [1.91] | [2.20] | [2.13] |
| Constant | -0.76 | -0.83 | -0.71 | -0.73 | -0.81 | -0.24 | -0.86 | -1.28 |
|  | [0.85] | [0.85] | [0.86] | [0.86] | [0.86] | [0.76] | [0.90] | [1.08] |
| *Dummy* | Island Countries | SARS Countries | Max Two Neighbour Countries | High Trade Countries | Elimination Countries plus ESP, TUR, IDN | Elimination Countries | Elimination Countries | Elimination Countries |
| Time Fixed Effects (TFE) | Q1 2014 –Q4 2020 (pre-2020 quarters not displayed) | | | | | | | |
| Sample | All | All | All | All | All | OECD | Excl. CHN | Bigger 3m |
| Model | FE | FE | FE | FE | FE | FE | FE | FE |
| Countries | 44 | 44 | 44 | 44 | 44 | 37 | 43 | 38 |
| R2 (adj.) | 0.71 | 0.71 | 0.71 | 0.71 | 0.71 | 0.72 | 0.71 | 0.71 |
| R2 (within) | 0.71 | 0.71 | 0.71 | 0.71 | 0.71 | 0.72 | 0.72 | 0.72 |
| R2 (overall) | 0.53 | 0.53 | 0.53 | 0.53 | 0.53 | 0.59 | 0.56 | 0.52 |
| R2 (between) | 0.04 | 0.04 | 0.03 | 0.03 | 0.03 | 0.01 | 0.01 | 0.06 |
| Rho (inter. cor.) | 0.53 | 0.54 | 0.53 | 0.53 | 0.53 | 0.44 | 0.50 | 0.56 |
| F-Statistic | 159.08 | 147.26 | 153.24 | 153.37 | 244.38 | 155.37 | 193.57 | 725.92 |

Note: *Island Countries* (n = 7) in (1) are AUS, GBR, IDN, IRL, ISL, JPN, NZL. *SARS Countries* (n = 18) in (2) are AUS, CAN, CHE, CHN, DEU, ESP, FRA, GBR, IDN, IND, IRL, ITA, KOR, NZL, RUS, SWE, USA, ZAF. *Two Neighbours Max* (n = 14) in (3) are AUS, CAN, DNK, EST, GBR, IRL, ISL, JPN, KOR, NLD, NZL, PRT, SWE, USA, *High Trade Countries* (n = 22) in (4) are AUT, BEL, CHE, CZE, DEU, DNK, EST, FIN, HUN, IRL, ISL, KOR, LTU, LUX, LVA, MEX, NLD, POL, PRT, SVK, SVN, SWE. Non-OECD countries (n = 6) excluding in (8) are ARG, BRA, CHN, IDN, IND, RUS, ZAF. Countries with less than 3m inhabitants (n = 6) excluding in (10) are EST, ISL, LTU, LVA, LUX, SVN.

Overall, robustness checks do not provide unambiguous support for the claim that elimination strategy countries were able to reduce growth and health risks by implementing NPIs swiftly and strictly compared to countries following a suppression / mitigation strategy. However, we also do not find evidence for the opposite. No robustness check shows that by pursuing a suppression / mitigation strategy countries could significantly lower the economic costs associated with NPI changes compared to elimination countries.

## Conclusions

This paper addresses the question whether the 2020 growth performance of countries pursuing a COVID-19 elimination strategy as defined by [11] was significantly different from the performance in countries running a suppression / mitigation strategy. We do so by analyzing within a 44 country sample whether the effects of non-pharmaceutical interventions (NPIs) on quarterly GDP growth differed in elimination strategy countries from those recorded in suppression / mitigation strategy countries. After identifying five countries as elimination strategy countries, namely Australia, China, Japan, South Korea and New Zealand, our baseline result shows that these countries were able to implement NPIs at lower macroeconomic costs in terms of GDP growth than countries following the alternative suppression / mitigation strategy. This indicates that a swift and strong response to the virus did not only save lives but also paid off in macroeconomic terms: GDP growth is less affected by NPI changes than in countries which aim to "live with the virus" and hence respond slowly and in a gradual way to rising infections.

Having said this, caution is warranted when drawing strong policy conclusions from this result as probit regressions indicate that the governments of the elimination countries might have opted for the strategy due to exogenous country characteristics facilitating its implementation and lowering its costs. Moreover, the statistical significance of the interaction term coefficient in growth regressions is sensitive to sample composition. Thus, the strategy choice might not have been causal for the different growth impact of NPI changes. In particular, our results should not be misinterpreted as indicating that non-elimination countries would have fared better in economic terms if they had opted for the elimination strategy.

At the same time, we do not find any evidence suggesting that the countries we identify as elimination strategy countries performed worse in terms of growth than countries following a suppression / mitigation strategy. This is a striking result on its own given that elimination countries did significantly better in terms of health risk reduction by showing less infections and lower mortality associated with COVID-19 than suppression / mitigation countries. Thus, our analysis can be summarized as follows: we find mixed support for the view that countries pursuing an elimination strategy faced lower costs in terms of GDP growth decline when implementing NPIs. At the same time, the empirical evidence does not support the notion that countries fighting the virus by NPIs with the goal of virus elimination had to accept lower GDP growth compared to countries which aimed at balancing health and economic risks by trying to "live with" the virus. This indicates that the very countries which opted for consistently employing the elimination strategy benefited from this decision.

## Supporting information

**S1 File.**
(RAR)

## Acknowledgments

We thank Filippo Mazzocca for excellent research assistance as well as Holger Sandte and two anonymous referees for helpful comments and suggestions.

## Author Contributions

**Conceptualization:** Michael König, Adalbert Winkler.

**Data curation:** Michael König.

**Formal analysis:** Michael König, Adalbert Winkler.

**Investigation:** Michael König, Adalbert Winkler.

**Methodology:** Michael König, Adalbert Winkler.

**Project administration:** Adalbert Winkler.

**Resources:** Michael König, Adalbert Winkler.

**Software:** Michael König, Adalbert Winkler.

**Supervision:** Michael König, Adalbert Winkler.

**Validation:** Michael König, Adalbert Winkler.

**Visualization:** Michael König.

**Writing – original draft:** Michael König, Adalbert Winkler.

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
