## [Decision Letter · Decision Letter 0]

9 Jul 2021

PONE-D-21-12254

The impact of government responses to the COVID-19 pandemic on GDP growth - Does strategy matter?

PLOS ONE

Dear Dr. König,

Thank you for submitting your manuscript to PLOS ONE. After careful consideration, we feel that it has merit but does not fully meet PLOS ONE’s publication criteria as it currently stands. Therefore, we invite you to submit a revised version of the manuscript that addresses the points raised during the review process.

We look forward to receiving your revised manuscript.

Kind regards,

Bing Xue, Ph.D.

Academic Editor

PLOS ONE

Additional Editor Comments (if provided):

Reviewers' comments:

Reviewer's Responses to Questions

**Comments to the Author**

1. Is the manuscript technically sound, and do the data support the conclusions?

Reviewer #1: Yes

Reviewer #2: Yes

2. Has the statistical analysis been performed appropriately and rigorously? 

Reviewer #1: Yes

Reviewer #2: Yes

3. Have the authors made all data underlying the findings in their manuscript fully available?

Reviewer #1: Yes

Reviewer #2: Yes

4. Is the manuscript presented in an intelligible fashion and written in standard English?

Reviewer #1: Yes

Reviewer #2: Yes

5. Review Comments to the Author

Reviewer #1: Referee Report – “The impact of government responses to the COVID-19 pandemic on GDP growth – Does strategy matter?”

Summary. This paper studies whether countries, which decided to ‘eliminate’ COVID-19 instead of just mitigating its impact on national health, performed better with regard to national GDP development over time. Hence, this paper deals with a very important question for the current crisis and also may derive some implications for future pandemics. Their quantitative results are founded by detailed qualitative cross-checks. Though, I think the empirical strategy may be problematic with regard to a selection process into the group of ‘elimination’ countries.

Comments. Please find some remarks below which may help to improve your manuscript.

1. Endogeneity: I am a bit worried about your regression setup. It is highly endogenous to become part of the group of ‘eliminating’ states. You also argue this in your paper. While I am not really sure what do to about it, it remains an issue. You tackle the issue in providing several robustness checks etc. and hence already provide some insight that the regressions are not biased. Still, it remains a problem and you should argue that you analyze no causal, but descriptive findings.

2. It might be a problem to use the growth rate as dependent variable as it is serially correlated over time within a country. If I have a strong drop in the first quarter, I typically see a stronger increase in the next quarter. So, may it be better to use absolute GDP and country FE?

3. Note that stringency is also related to factors like the quality of the national health system etc. How may this be captured in your regressions?

4. In the early parts of your paper, you use the reaction at an incidence level of 5 (1) as relative threshold for studying states’ behavior. I know that it is hard to capture the differences in the testing density across state, but this may definitely affect such a threshold. Also, countries with a worse health system may react early. Can you motivate whether this may affect your intuition?

5. I think you are right in saying that you cannot clearly derive policy implications from your findings as countries strongly differ in their exogenous ability to ‘eliminate’ COVID-19. But can you maybe say something about whether you think countries who followed the ‘elimination’ strategy did this correctly?

Reviewer #2: - Spelling and grammar is okay throughout the paper.

- The paper should preferably be in "past-tense." Some changes are needed in this regard.

- The "instrument" or "method" used for data collection or the "data set" used for the qualitative analysis is unclear.

- The analysis sections could be restructured. All the quantitative analysis parts could be group under one section and all the qualitative analysis parts under another. This will make the paper easier to understand.

- The reference style is okay and used with confidence.

Overall, the paper is well written. It will be ready for publication if the minor points mentioned are addressed.

6. PLOS authors have the option to publish the peer review history of their article (what does this mean?). If published, this will include your full peer review and any attached files.

Reviewer #1: No

Reviewer #2: No

---

## [Author Response · Author response to Decision Letter 0]

26 Aug 2021

We would like to thank the reviewers for their helpful comments. PLEASE EXCUSE THE DIFFICULTY TO READ THE TABLES. IN THIS RESPONSE FORM TABLES CAN NOT BE INCLUDED. 

Detailed response to the reviewer comments (in italics)

Reviewer #1: Referee Report - "The impact of government re-sponses to the COVID-19 pandemic on GDP growth - Does strategy matter?"

Summary. This paper studies whether countries, which decided to 'eliminate' COVID-19 instead of just mitigating its impact on national health, performed better with regard to national GDP development over time. Hence, this paper deals with a very important question for the current crisis and also may derive some implications for fu-ture pandemics. Their quantitative results are founded by detailed qualitative cross-checks. Though, I think the empirical strategy may be problematic with regard to a selection process into the group of 'elimination' countries.

Comments. Please find some remarks below which may help to im-prove your manuscript.

1. Endogeneity: I am a bit worried about your regression setup. It is highly endogenous to become part of the group of 'eliminating' states.

You also argue this in your paper. While I am not really sure what do to about it, it remains an issue. You tackle the issue in providing sev-eral robustness checks etc. and hence already provide some insight that the regressions are not biased. Still, it remains a problem and you should argue that you analyze no causal, but descriptive find-ings.

Our Response: We follow this suggestion and modify the language accordingly. Moreover, at the end of the introduction we explicitly follow up on this comment by inserting that “WE DO NOT CLAIM THAT RUNNING AN ELIMINATION STRATEGY WAS CAUSAL FOR MINIMIZING HEALTH RISKS AT LOWER ECONOMIC COSTS AS IT IS UNCLEAR WHETHER THE ADOPTION OF SUCH A STRATEGY IN OTHER COUNTRIES WOULD HAVE MADE NPIS LESS COSTLY IN TERMS OF GROWTH COMPARED TO THE OBSERVED SUPPRESSION / MITIGATION STRATEGY OUTCOME.” In the final section we close the paragraph calling for caution when drawing policy conclusions with the following sen-tences: “THUS, THE STRATEGY CHOICE MIGHT NOT HAVE BEEN CAUSAL FOR THE DIFFERENT GROWTH IMPACT OF NPI CHANGES. IN PARTICULAR, OUR RESULTS SHOULD NOT BE MISINTERPRETED AS INDICATING THAT NON-ELIMINATION COUN-TRIES WOULD HAVE FARED BETTER IN ECONOMIC TERMS IF THEY HAD OPTED FOR THE ELIMINATION STRATEGY.” 

2. It might be a problem to use the growth rate as dependent varia-ble as it is serially correlated over time within a country. If I have a strong drop in the first quarter, I typically see a stronger increase in the next quarter. So, may it be better to use absolute GDP and coun-try FE?

Our Response: Serial correlation is an important issue. Moreover, as it is pointed out, during the COVID-19 pandemic the growth pat-tern is such that a deeper drop in the previous quarter tends to be associated with a lower drop / stronger rise in the current quarter in line with the “hammer and dance” view of NPI policies. 

Having said this, the evidence – for example for the US – suggests that GDP growth exhibits a mild positive autocorrelation: if GDP grows faster than average in one period, there is a tendency for it to grow faster than average in the following periods [Following Cogley and Nason (1995), this holds for real US GNP growth in the short run. In the longer run, i.e. at higher lags, the autocorrelations are mostly negative and statistically insignificant]. As our observa-tion period covers six pre-COVID-19 years (starting in Q1 2014 until Q4 2019, representing 24 pre-COVID-19 quarters), we believe that GDP growth is the preferred dependent variable.

Nevertheless, we followed the suggestion running the analysis with GDP levels instead of GDP growth as a dependent variable. We met the challenge for expressing GDP in the same dimension for all countries under review by indexing real GDP in local currency (data expressed in the same currency, i.e. USD, cannot be used as this would imply that exchange rate changes influence GDP develop-ments, i.e. cross-country real GDP level differences over time would not only reflect different paths of real output growth but also ex-change rate developments). 

GDP levels broadly follow an upward trend. Thus, running fixed ef-fects regressions, where variables are basically demeaned, implies that most variation in GDP levels is captured by time fixed effects. Time periods before GDP levels reach the coefficient for the con-stant (1.17 in the Table R1 below) show (declining) negative coeffi-cients (i.e. in the regression the reference year is Q4 2019. As sug-gested the time fixed effects in 2014 are more negative, than in 2015 and so on).

For the COVID-19 period results show that the stringency index has no significant influence on GDP levels over time (see below). This also holds for the stringency index lagged and the interaction term between the elimination country dummy and stringency. However, quarterly COVID-19 deaths per 100,000 people have a significantly negative impact. As elimination countries have very low death numbers, the results are materially in line with those recorded in the paper, but are triggered by different transmission mechanisms: in the growth regression countries successfully eliminating COVID-19 via NPIs reap economic benefits by recording lower declines in real GDP growth when raising NPI stringency, in the GDP level re-gression elimination countries benefit economically compared to their non-elimination peers as they record very low death numbers. 

Still, we believe that the level regression is the one less emphasis should be placed upon as the deep recession in the second quarter 2020 is captured by the time fixed effect for the second quarter. Thus, the negative impact of a rising stringency index on output is likely hidden in the time fixed effect. While this might also hold – at least partly – for the growth regression reported in the paper, the dimension is much larger in the level than in the growth regression. 

Table R1: GDP level (index, 2013=100), the stringency index and COVID-19 fatality - fixed effects regressions (modification of Table 5 in the paper)

Dependent variable 

GDP level (index) (1) (2) (3)

Stringency -0.0001 -0.0005 -0.0007

 [0.0007] [0.0005] [0.0006]

Stringency (lag) - 0.0011 0.0010

 - [0.0010] [0.0009]

Elimination x Stringency - - 0.0013

 - - [0.0010]

Fatality -0.0013** -0.0012** -0.0008**

 [0.0006] [0.0005] [0.0004]

Share 0.0332 0.0350 0.0329

 [0.0246] [0.0248] [0.0248]

2014_1 -0.1819*** -0.1817*** -0.1820***

 [0.0255] [0.0254] [0.0256]

2014_2 -0.1756*** -0.1753*** -0.1756***

 [0.0231] [0.0230] [0.0231]

2014_3 -0.1666*** -0.1664*** -0.1667***

 [0.0222] [0.0221] [0.0222]

2014_4 -0.1555*** -0.1553*** -0.1556***

 [0.0205] [0.0204] [0.0205]

2015_1 -0.1504*** -0.1503*** -0.1505***

 [0.0208] [0.0207] [0.0209]

2015_2 -0.1426*** -0.1425*** -0.1426***

 [0.0184] [0.0183] [0.0184]

2015_3 -0.1333*** -0.1331*** -0.1333***

 [0.0173] [0.0172] [0.0174]

2015_4 -0.1246*** -0.1244*** -0.1247***

 [0.0147] [0.0146] [0.0148]

2016_1 -0.1243*** -0.1240*** -0.1244***

 [0.0194] [0.0192] [0.0194]

2016_2 -0.1144*** -0.1142*** -0.1145***

 [0.0160] [0.0159] [0.0161]

2016_3 -0.1083*** -0.1081*** -0.1083***

 [0.0152] [0.0151] [0.0152]

2016_4 -0.0913*** -0.0912*** -0.0914***

 [0.0111] [0.0111] [0.0112]

2017_1 -0.0929*** -0.0927*** -0.0929***

 [0.0154] [0.0153] [0.0154]

2017_2 -0.0793*** -0.0793*** -0.0793***

 [0.0119] [0.0118] [0.0119]

2017_3 -0.0700*** -0.0700*** -0.0700***

 [0.0102] [0.0103] [0.0103]

2017_4 -0.0554*** -0.0554*** -0.0554***

 [0.0071] [0.0071] [0.0071]

2018_1 -0.0542*** -0.0543*** -0.0542***

 [0.0111] [0.0112] [0.0111]

2018_2 -0.0429*** -0.0429*** -0.0429***

 [0.0077] [0.0077] [0.0077]

2018_3 -0.0358*** -0.0358*** -0.0358***

 [0.0065] [0.0065] [0.0065]

2018_4 -0.0236*** -0.0235*** -0.0236***

 [0.0046] [0.0046] [0.0046]

2019_1 -0.0230** -0.0230** -0.0230**

 [0.0094] [0.0094] [0.0094]

2019_2 -0.0119** -0.0119** -0.0119**

 [0.0058] [0.0058] [0.0058]

2019_3 -0.0061 -0.0061 -0.0061

 [0.0043] [0.0043] [0.0043]

2019_4 0.0000 0.0000 0.0000

 [.] [.] [.]

2020_1 -0.0274 -0.0185 -0.0207

 [0.0207] [0.0143] [0.0137]

2020_2 -0.1120*** -0.1047*** -0.1053***

 [0.0407] [0.0382] [0.0381]

2020_3 -0.0281 -0.0783 -0.0736

 [0.0326] [0.0746] [0.0639]

2020_4 0.0234 -0.0120 -0.0159

 [0.0312] [0.0545] [0.0516]

Constant 1.1756*** 1.1737*** 1.1760***

 [0.0317] [0.0314] [0.0321]

Model FE FE FE

Countries 44.00 44.00 44.00

R2 (adj.) 0.53 0.53 0.54

R2 (within) 0.54 0.54 0.55

R2 (overall) 0.16 0.16 0.17

R2 (between) 0.05 0.05 0.03

Rho (inter. cor.) 0.81 0.81 0.81

F-Stat. 67.53 55.02 66.83

3. Note that stringency is also related to factors like the quality of the national health system etc. How may this be captured in your regressions?

Our Response: We agree that the stringency level does not only depend on the “exogenous” shock captured by the pandemic itself, but also on other variables. In the Probit regression we explicitly account for some of these factors as we assess which factors influ-ence a country’s decision to employ NPIs strictly and swiftly with the goal of eliminating the virus. 

Thus, as a first response to the comment, we add the Global Health Security Index (https://www.ghs¬index.¬org/) as reported for 2019 to the list of explanatory variables in the Probit regression. Results (see Table R2 below) show that the index has no explanatory power. This also holds when we drop the government effectiveness index as an explanatory variable (the GHS index and the government ef-fectiveness show a high degree of correlation (0.556***, *** p < 0.001) in our 44 country sample).

This is in line with other studies (see Abbey et al. 2020 (https://doi.org/10.1371/journal.pone.0239398)) indicating that there is no relationship between the pre-pandemic quality of the health system, for example as measured by the GHS index, and the health performance of countries during the pandemic.

Table R2: Probit regression (average marginal effects) – including the Global Health Security Index (modification of Table 3 in the pa-per)

Dependent variable: 

Elimination Strategy (1) (2) (3) (4) (5) (6)

SARS 0.124 0.128 0.100 

 (0.094) (0.092) (0.079) 

 [-0.060; 0.308] [-0.051; 0.308] [-0.055; 0.255] 

Island 0.177** 0.167** 0.119* 0.130** 0.119* 0.176***

 (0.083) (0.080) (0.067) (0.065) (0.069) (0.067)

 [0.014; 0.340] [0.011; 0.324] [-0.012; 0.250] [0.002; 0.257] [-0.016; 0.254] [0.045; 0.307]

Government 0.034 0.076 0.083* 0.113* 

 (0.053) (0.048) (0.043) (0.062) 

 [-0.069; 0.138] [-0.019; 0.171] [-0.002; 0.168] [-0.009; 0.235] 

Trade -0.003 -0.003* -0.003 -0.002*

 (0.002) (0.002) (0.002) (0.001)

 [-0.006; 0.001] [-0.007; 0.001] [-0.008; 0.001] [-0.005; 0.000]

Health_Overall -0.002 0.001

 (0.005) (0.004)

 [-0.011; 0.006] [-0.006; 0.008]

Countries 44 44 44 44 44 44

Pseudo R² 0.26 0.27 0.40 0.36 0.37 0.37

Note: Binary Probit model. *,**,*** denote significance at 10, 5, and 1 percent levels, respectively. Values represent marginal effects (dy/dx). Values in parenthesis repre-sent robust standard errors; 95%-confidence interval of dy/dx is in squared brackets.

Accordingly, we insert the following footnote 17: “WE ALSO RUN RE-GRESSIONS INCLUDING THE GLOBAL HEALTH SECURITY INDEX (HTTPS://WWW.GHSINDEX.ORG/) AS REPORTED FOR 2019 TO THE LIST OF EX-PLANATORY VARIABLES IN ORDER TO FOCUS MORE ON THE QUALITY OF HEALTH SYS-TEMS AND THE PREPAREDNESS OF COUNTRIES IN DEALING WITH A PANDEMIC. HOWEVER, THE INDEX HAS NO EXPLANATORY POWER. THIS ALSO HOLDS WHEN WE DROP THE GOVERNMENT EFFECTIVENESS INDEX AS AN EXPLANATORY VARIABLE. OUR RESULTS ARE IN LINE WITH OTHER STUDIES ([35] ABBEY ET AL. 2020) INDI-CATING THAT THERE IS NO RELATIONSHIP BETWEEN THE PRE-PANDEMIC QUALITY OF THE HEALTH SYSTEM, AS MEASURED BY THE GHS INDEX, AND THE HEALTH PERFOR-MANCE OF COUNTRIES DURING THE PANDEMIC.”

A second and more direct option for testing the impact of the quali-ty of national health systems on the stringency index would be run-ning regressions reported in Table 5 as instrumental variable re-gressions, i.e. instrumenting the stringency index by health system quality, for example as measured by the GHS index. However, as we run fixed effects regression with quarterly data the instrument also has to have quarterly frequency, i.e. has to be time-variant. This precludes making use of the GHS index. Moreover, we are not aware of any data capturing health system quality as such on a quarterly basis (and even if quarterly data existed, changes would likely be minimal and hence the instrument quality for stringency would likely be low). Thus, we are unable to pursue this option. 

4. In the early parts of your paper, you use the reaction at an inci-dence level of 5 (1) as relative threshold for studying states' behav-ior. I know that it is hard to capture the differences in the testing density across state, but this may definitely affect such a threshold. Also, countries with a worse health system may react early.

Can you motivate whether this may affect your intuition?

Our Response: In the answer to 3) we have already responded to the question whether health system quality affects timing and strength of the NPI response to changes in cases. Thus, our answer focuses on the relationship between testing and cases. 

We agree that the definition of a threshold requires that the num-ber of cases reported in the Oxford University database is close to the “true” number. We also agree that the number of cases report-ed might be driven by the number of tests conducted, if countries have strongly different testing intensities. 

Our world in data (https://ourworldindata.org/coronavirus-testing) reports the number of tests conducted per 1,000 inhabit-ants over time. Based on this we calculated quarterly testing activi-ties per 1,000 inhabitants for the four quarters in 2020. As ex-pected, results show that on average the number of tests has strongly risen over time for the countries of our sample (Table R3). Moreover, results also show that the elimination countries test sub-stantially less than non-elimination countries. Indeed, in Q4 the number of tests in the former is about 50 times larger than in the latter. This raises concerns that the low incidence rate reported for the elimination countries might be “artificially low”, i.e. the “true” incidence rate in these countries might be substantially higher than threshold of 5.

Table R3: Number of COVID-19 tests (per 1,000 inhabitants) 

Tests Q1 Q2 Q3 Q4

Total 56,31 219,82 449,56 2154,32

Elimination countries 13,21 10,21 31,31 51,41

Non-elimination countries 62,47 248,93 507,65 2394,65

Sources: Our World in Data, authors’ calculations

However, the number of tests is endogenous: it does not make sense to test large parts of the population without any evidence that people might be infected. In countries successfully eliminating the virus this is the case.

Against this background, it is useful to look at the test positive rate. It “is a good metric for how adequately countries are testing be-cause it indicates the level of testing relative to the size of the out-break. To be able to properly monitor and control the spread of the virus, countries with more widespread outbreaks need to do more testing. According to criteria published by WHO in May 2020, a positive rate of less than 5% is one indicator that the epidemic is under control in a country. Because limited testing makes it likely that many cases will be missed, the positive rate can also help our understanding of the spread of the virus. In countries with a high positive rate, the number of confirmed cases is likely to represent only a small fraction of the true number of infections. And where the positive rate is rising in a country, this can suggest the virus is actually spreading faster than the growth seen in confirmed cases.” (https://ourworldindata.org/coronavirus-testing#the-positive-rate-a-crucial-metric-for-understanding-the-pandemic).

Below we report test positive rate means for the countries of our sample over the four quarters (Table R4). Results strongly mitigate concerns that the “true” incidence rate in elimination countries might be much higher than reported as

a) the positive rate is substantially lower than in non-elimination countries;

b) the positive rate is always below the 5 % benchmark set by the WHO in elimination but above the 5% benchmark in the non-elimination countries.

c) while positive rates co-move over time, only from Q2 to Q3 there is a small rise in the rate for the elimination, but a small decline in the non-elimination countries, the positive rate in the non-elimination countries is more than 6 times higher than in the elimi-nation countries starting with Q2 2020.

Table R4: Test positive rates in 2020: total sample, elimination and non-elimination countries 

Test positive rate Q1 Q2 Q3 Q4

Total 8,96% 7,35% 7,49% 11,32%

Elimination countries 3,59% 1,06% 1,16% 2,03%

Non-elimination count-ries 9,73% 7,79% 7,62% 12,38%

Sources: Our World in Data, authors’ calculations

The testing regime index (ranging from 0 to 3, with a higher num-ber indicating a tighter regime) compiled by Oxford University (Hale et al. 2020) also suggests that the strictness of testing regimes does not significantly differ between elimination and non-elimination countries (Table R5.

We conclude from this that the low incidence levels recorded in elimination countries reflect “true” levels despite the number of tests being substantially lower in elimination than in non-elimination countries. Hence, we have confidence that they are be-low the IR threshold of 5 set in the paper in the elimination and above that threshold in the non-elimination countries.

Table R5: Two-sample T-tests (unequal variances) – testing regimes 

Group Observations Mean Standard Deviation

Non-elimination coun-tries 156 1.72 0.84

Elimination countries 20 1.92 0.19

Combined 176 1.74 0.06

Diff -0.20 0.20

Diff = mean (Non-Elimination countries) – mean (Elimination countries) t=-1.02

H0: diff = 0 Satterthwaite’s degrees of freedom 174

H1: diff < 0 H1: diff != 0 H1: diff > 0

Pr(T < t) = 0.1542 Pr(|T |> |t |= 0.3084 Pr(T > t) = 0.8458

Note: Based on four quarters in 2020. Unequal variances based on variance ratio test Variable of inter-est is Fatality. T-test with unequal variances shows significant difference between Elimination and Non-Elimination countries. Testing Policy is an ordinal structured variable with four characteristics. 0 – No testing policy; 1 – Only those who both (a) have symptoms AND (b) meet specific criteria (e.g. key workers, admitted to hospital, came into contact with a known case, returned from overseas); 2 – testing of anyone showing COVID-19 symptoms; 3 – open public testing (e.g.“drive through” testing available to asymptomatic people).

We perform an additional test by exploiting the fact that the corre-lation between tests and deaths is substantially lower than between tests and cases (Table R6). Concretely, we recalculate Table 2 by replacing the incidence rate recorded at the strongest rise (lock-downs 1 and 2) and fall (opening 1) with the death rate. 

Table R6: Correlation coefficients – COVID-19 tests, cases and relat-ed deaths, quarterly data, Q1-Q4 2020 (44 countries) 

 Tests Cases Deaths

Total_Tests (relative to population) 1 

Cases (relative to population) 0.594*** 1 

Deaths (relative to population) 0.293*** 0.786*** 1

* p < 0.05, ** p < 0.01, *** p < 0.001

Results (Table R7) again show that China, South Korea, Japan, Aus-tralia and New Zealand respond at significantly lower COVID-19 related death rates with the strongest rise (fall) of the stringency index during the first wave and the beginning of the second wave compared to the remaining countries. 

Table R7: Waves, stringency on non-pharmaceutical interventions, incidence and death rates – country ranking (rank 1 – 10), expan-sion and modification of Table A1 in the previous version

Rank Coun-try Average COVID-19 related fatality rate over the 7-day period prior to the maximum change in the strin-gency index (lockdown 1 and 2, Re-Opening 1) Coun-try Incidence rate over the 7-day period prior to the maximum change in the stringency index (lockdown 1 and 2, Re-Opening 1)

(-> Incidence rate used in the paper, see S1 Table for full country overview)

 Fatality Incidence rate

1 NZL 0,00 CHN 0,07

2 CHN 0,01 KOR 0,27

3 AUS 0,02 JPN 1,49

4 KOR 0,04 AUS 1,5

5 JPN 0,05 NZL 1,57

6 NOR 0,13 ESP 7,18

7 FIN 0,42 TUR 7,35

8 AUT 0,52 IDN 7,69

9 IDN 0,53 BEL 14,96

10 ISL 0,59 ZAF 15,04

Note: Value represents the mean fatality rate in the time of Lock-down 1, Re-Opening 1, and Lockdown 2. The lower the value the earlier (later) a government enacted maximum changes in SI facing a rise (decline) in the 7-day fatality rate. 

We report on these findings in the revised version of the paper by introducing a footnote (footnote 11) which reads as follows: “BY SETTING A COMMON BENCHMARK BASED ON REPORTED CASES, WE RUN THE RISK THAT COUNTRIES MIGHT QUALIFY AS ELIMINATION COUNTRIES BECAUSE THEY TEST LESS EXTENSIVELY THAN OTHER COUNTRIES AND HENCE REPORT FEWER CASES. AT A FIRST GLANCE, CROSS-COUNTRY DATA ON COVID-19 TESTING PROVIDED BY [19 = HASELL ET AL. (2020)] POINT IN THIS DIRECTION AS THE COUNTRIES REPORTING INCIDENCE RATES BELOW 5 TEST SIGNIFICANTLY LESS THAN THEIR PEERS. HOWEVER, THEY ALSO SHOW A MUCH LOWER TEST POSITIVE RATE THAN THEIR PEERS WHICH IS CONSISTENTLY BELOW THE 5% BENCHMARK SET BY THE WHO FOR CATEGORIZING COUNTRIES AS HAVING THE PANDEMIC UNDER CONTROL. MOREOVER, WHEN RE-PLACING THE 7-DAY INCIDENCE RATE WITH A 7-DAY MOVING AVERAGE FOR COVID-19 RELATED DEATHS WE IDENTIFY EXACTLY THE SAME COUNTRIES AS ELIM-INATION STRATEGY COUNTRIES EVEN THOUGH THE CORRELATION BETWEEN INTEN-SITY OF COVID-19 TESTS AND RELATED DEATHS IS SUBSTANTIALLY SMALLER THAN BETWEEN COVID-19 TESTS AND NUMBER OF CASES. THUS, WE ARE CONFIDENT THAT DIFFERENCES IN TESTING DENSITY ACROSS COUNTRIES DO NOT DRIVE OUR RESULTS. WE THANK AN ANONYMOUS REVIEWER FOR ALERTING US TO THIS ISSUE.”

5. I think you are right in saying that you cannot clearly derive policy implications from your findings as countries strongly differ in their exogenous ability to 'eliminate' COVID-19. But can you maybe say something about whether you think countries who followed the 'elimination' strategy did this correctly?

Our Response: We follow up on this comment by inserting the fol-lowing sentence already in the abstract: “AT THE SAME TIME OUR RE-SULTS SHOW THAT COUNTRIES SUCCESSFULLY APPLYING THE ELIMINATION STRATE-GY ACHIEVED BETTER HEALTH OUTCOMES THAN THEIR PEERS WITHOUT HAVING TO ACCEPT LOWER GROWTH.” We repeat this sentence in somewhat modi-fied forms at the end of the introduction and the end of the con-cluding section.

 

Reviewer #2: - Spelling and grammar is okay throughout the paper.

- The paper should preferably be in "past-tense." Some changes are needed in this regard.

Our Response: We follow up on this comment and make use of the past-tense when appropriate and needed.

- The "instrument" or "method" used for data collection or the "da-ta set" used for the qualitative analysis is unclear.

Our Response: We revise the introduction into the qualitative anal-ysis in the main text (footnote 15) and in the S1 File (S2). Concrete-ly, we clarify that the data set used is the same applied in the quan-titative analysis. Moreover, we spell out that the analysis relies on a visual inspection of plots with the goal of identifying episodes where NPI policies are clearly inconsistent with the strategy princi-ples, in particular the principles on which the elimination strategy is built upon, listed on page 5. 

- The analysis sections could be restructured. All the quantitative analysis parts could be group under one section and all the qualita-tive analysis parts under another. This will make the paper easier to understand.

Our Response: We follow up on this suggestion by focusing on the quantitative analysis in the main text while relegating the introduc-tion into the qualitative analysis, including the illustration with the case of Argentina, to the table in the annex. 

- The reference style is okay and used with confidence.

Overall, the paper is well written. It will be ready for publication if the minor points mentioned are addressed.

 

References:

Abbey, E. J., Khalifa, B. A., Oduwole, M. O., Ayeh, S. K., Nudotor, R. D., Salia, E. L., Lasisi, O., Bennett, S., Yusuf, H.E., Agwu, A.L., Kara-kousis, P. C. (2020). The Global Health Security Index is not predic-tive of coronavirus pandemic responses among Organization for Economic Cooperation and Development countries. PloS one, 15(10), e0239398.

Cogley, T., Nason, J. M. (1995). Output dynamics in real-business-cycle models. The American Economic Review, 492-511.

Hale, T., Noam A., Beatriz K., Anna P., Toby P., Samuel W. (2020). Variation in Government Responses to COVID-19. BSG Working Pa-per Series. BSG-WP-2020/032 Version 6.0. www.bsg.ox.ac.uk/covidtracker. 

Hasell, J., Mathieu, E., Beltekian, D., Macdonald, B., Giattino, C., Ortiz-Ospina, E., Roser, M, Ritchie, H. (2020). A cross-country data-base of COVID-19 testing. Scientific data, 7(1), 1-7.

---

## [Decision Letter · Decision Letter 1]

19 Oct 2021

The impact of government responses to the COVID-19 pandemic on GDP growth - Does strategy matter?

PONE-D-21-12254R1

Dear Dr. König,

We’re pleased to inform you that your manuscript has been judged scientifically suitable for publication and will be formally accepted for publication once it meets all outstanding technical requirements.

Kind regards,

Bing Xue, Ph.D.

Academic Editor

PLOS ONE

Additional Editor Comments (optional):

Reviewers' comments:

Reviewer's Responses to Questions

**Comments to the Author**

1. If the authors have adequately addressed your comments raised in a previous round of review and you feel that this manuscript is now acceptable for publication, you may indicate that here to bypass the “Comments to the Author” section, enter your conflict of interest statement in the “Confidential to Editor” section, and submit your "Accept" recommendation.

Reviewer #1: All comments have been addressed

Reviewer #2: All comments have been addressed

2. Is the manuscript technically sound, and do the data support the conclusions?

Reviewer #1: Yes

Reviewer #2: Yes

3. Has the statistical analysis been performed appropriately and rigorously? 

Reviewer #1: Yes

Reviewer #2: Yes

4. Have the authors made all data underlying the findings in their manuscript fully available?

Reviewer #1: Yes

Reviewer #2: Yes

5. Is the manuscript presented in an intelligible fashion and written in standard English?

Reviewer #1: Yes

Reviewer #2: Yes

6. Review Comments to the Author

Reviewer #1: The authors addressed most of the points I mentioned in the referee report. Overall, their paper now mentions remaining downsides or improved.

Reviewer #2: This is an interesting study that demonstrating how strategies employed by governments to fight the COVID-19 pandemic affected GDP growth in 2020. This indicates that a clear response to COVID-19 not only saved lives but also led to economic rewards and showed GDP growth is lass affected by NPI changes. The study found that countries, which moved forward with suppression/mitigation strategies were less effective achieving GDP growth. However, since COVID-19 is a new phenomenon, the authors note that their findings should not be generalized.

7. PLOS authors have the option to publish the peer review history of their article (what does this mean?). If published, this will include your full peer review and any attached files.

Reviewer #1: No

Reviewer #2: No

---

## [Editor Report · Acceptance letter]

25 Oct 2021

PONE-D-21-12254R1 

The impact of government responses to the COVID-19 pandemic on GDP growth - Does strategy matter? 

Dear Dr. König:

I'm pleased to inform you that your manuscript has been deemed suitable for publication in PLOS ONE. Congratulations! Your manuscript is now with our production department. 

Kind regards, 

on behalf of

Professor Bing Xue 

Academic Editor

PLOS ONE